Treatment of the terrible triad of the elbow by olecranon osteotomy: a retrospective cohort study

Zhou Ming
Xue Yuan
Jia Xueyuan
Wang Jianbing
Wu Yongwei
Ma Yunhong
Sun Zhenzhong sunzhenzhong_spine@163.com
Rui Yongjun ryjwx_trauma@163.com
Department of Orthopaedics, Wuxi Ninth People’s Hospital Affiliated to Soochow University , Wuxi , China
Wang Jincheng
Electronic publication date: 2024 Nov 15
Publication date: 2024
Volume: 12
Electronic Location ID: e18469
Received 2024 Jul 8; Accepted 2024 Oct 15
Copyright: ©2024 Zhou et al.
Copyright year: 2024
Copyright holder: Zhou et al.
License: This is an open access article distributed under the terms of the Creative Commons Attribution License, which permits unrestricted use, distribution, reproduction and adaptation in any medium and for any purpose provided that it is properly attributed. For attribution, the original author(s), title, publication source (PeerJ) and either DOI or URL of the article must be cited.
License URL: https://creativecommons.org/licenses/by/4.0/

Keywords: Olecranon osteotomy, Terrible triad of the elbow, Elbow joint

Funding: Wuxi Top Medical Expert Team of “Taihu Talent Program” TTPJY2021 This study was provided by Wuxi Top Medical Expert Team of “Taihu Talent Program” (TTPJY2021). The funders had no role in study design, data collection and analysis, decision to publish, or preparation of the manuscript.

==============================
Background

This study aims to evaluate the surgical techniques and outcomes of treating elbow dislocations with coronoid and radial head fractures, commonly referred to as the terrible triad of the elbow (TTE), through a single olecranon osteotomy.

Methods

A retrospective analysis was performed on 73 patients diagnosed with TTE between January 2015 and April 2022. The cohort included 44 men and 29 women, with an average age of 40.0 ± 15.1 years (range, 18–68 years). Among these patients, Mason Type I, II, and III fractures were identified in 11, 42, and 20 cases, respectively, while Morrey Type I, II, and III fractures were observed in 45, 23, and five cases, respectively. All patients underwent treatment via a single olecranon osteotomy. The average interval between injury and surgery was 5.6 ± 1.6 days (range, 3–8 days). Elbow function was assessed using the Mayo Elbow Performance Score (MEPS), pain was measured via the Visual Analogue Scale (VAS), and quality of life was evaluated using the SF-36 questionnaire.

Results

Patients were followed for 15 to 60 months (mean, 37.1 ± 13.3 months). All coronoid and radial head fractures achieved complete healing, with an average recovery time of 4.3 ± 1.1 months (range, 3–6 months). By the final follow-up, all patients had regained normal elbow function. The mean elbow flexion was 124.4° ± 9.2°, extension was 9.6° ± 6.5°, and the total range of flexion-extension was 114.8° ± 11.7°. Forearm pronation averaged 77.3° ± 4.8°, supination 79.2° ± 6.5°, and total forearm rotation 156.5° ± 8.4°. The mean MEPS was 89.3 ± 6.4, with 36 patients achieving excellent scores and 37 obtaining good scores. Preoperative VAS scores averaged 8.78 ± 1.11, which significantly dropped to 0.97 ± 0.83 at the final follow-up (p = 0.000). The SF-36 preoperative PCS and MCS scores were 45.77 ± 3.59 and 60.67 ± 3.91, respectively, with postoperative improvements to 93.85 ± 2.65 (p = 0.000) and 95.79 ± 3.11 (p = 0.000).

Conclusion

This retrospective analysis indicates that a single olecranon osteotomy could be a viable treatment option for TTE. However, additional research involving a control group is essential to substantiate the efficacy of this technique.

Introduction

Elbow dislocation combined with coronoid and radial head fractures, commonly referred to as the terrible triad of the elbow (TTE), was first identified by Hotchkiss in 1996 (Hotchkiss, 1996). TTE is typically characterized by significant posterolateral rotational instability and is often associated with extensive soft tissue damage, particularly affecting the lateral collateral ligament (LCL) complex. The management of this intricate injury poses a substantial challenge for orthopedic surgeons, as it can result in severe complications such as internal fixation failure, elbow stiffness, and ulnar neuropathy. In the early 20th century, treatment approaches were largely conservative, leading to poor outcomes. Patients frequently experience joint stiffness, instability, and progression to arthritis (Waterworth et al., 2023). However, advancements in surgical techniques, internal fixation devices, and a deeper understanding of injury mechanisms have improved prognosis, with success rates now ranging from 70% to 90% (Ahmed Kamel et al., 2024; Bozon et al., 2022).

Several surgical cutaneous approaches have been proposed for managing TTE, including the lateral, lateral combined medial, lateral combined anterior, anterior, and posterior approaches (Carroll & Morrissey, 2022; Chen & Bi, 2016; Hou et al., 2021; Meena et al., 2020; Ohl & Siboni, 2021). Despite these developments, no consensus has been reached on the most effective approach, and standardization remains a challenge (Zha et al., 2021). Surgeons often favor either a single lateral approach or a combined medial and lateral approach. However, significant soft tissue swelling can compromise visibility in the lateral approach. Conversely, while the combined approach may enhance exposure, it has been linked to increased intraoperative disruption of soft tissue around the elbow, potentially heightening the risk of postoperative stiffness, heterotopic ossification (HO), and ulnar nerve complications (Zha et al., 2021; Zhou et al., 2018). Additionally, the complexity of these procedures makes it difficult for less experienced surgeons to attain proficiency within a short period.

The olecranon osteotomy approach has been commonly applied in the surgical treatment of intercondylar comminuted humeral fractures (Wilson et al., 2021). This technique provides excellent intraoperative exposure and is relatively straightforward to perform. However, its application in the management of TTE remains underexplored, with limited studies addressing its use in this context. This study aims to retrospectively assess the clinical outcomes of patients with TTE who underwent surgical treatment using the olecranon osteotomy approach. By evaluating its effectiveness in managing this complex injury, the study seeks to address the current gap in knowledge. Specifically, we analyzed the clinical outcomes of 73 patients treated with the olecranon osteotomy approach between January 2015 and April 2022, with a minimum follow-up period of one year. The research hypothesizes that surgical management of TTE via the olecranon osteotomy approach results in favorable outcomes and reduced complications.

Materials & Methods

Clinical data

General information

Patients admitted to the Department of Orthopedic Surgery at Wuxi Ninth People’s Hospital with acute terrible triad injuries of the elbow, treated using the olecranon osteotomy approach between January 2015 and April 2022, were included in this study. The inclusion criteria were: (1) unilateral terrible triad confirmed by X-ray or CT scan; (2) fresh, closed fractures (≤2 weeks old); (3) age ≥ 18 years; (4) a follow-up period exceeding 12 months; and (5) treatment via the olecranon osteotomy approach. Exclusion criteria were: (1) incomplete clinical data; (2) bilateral terrible triad; (3) age < 18 years; (4) pathological fractures; (5) loss to follow-up or follow-up period < 1 year; (6) history of metabolic diseases or mental illness; (7) severe medical conditions precluding surgery; and (8) surgical delay of more than 2 weeks.

During the emergency phase, a closed reduction of the elbow was followed by temporary immobilization with a plaster slab. Surgery was scheduled after necessary medical assessments were completed. Of the 112 patients identified, 39 met the inclusion criteria after applying the exclusions, as shown in Fig. 1. This study received approval from the Ethics Committee of Wuxi Ninth People’s Hospital (NO: LW20220017), and all participants provided written informed consent. The study adhered to the ethical principles of the Declaration of Helsinki. All surgeries were conducted by a single team of senior orthopedic trauma surgeons with extensive experience in elbow joint procedures.

Figure 1 Study flow diagram.

Surgical methods

All patients were positioned either laterally or supine and received brachial plexus anesthesia. The affected limb was placed on a permeable table, and after standard disinfection, a balloon tourniquet was applied for hemostasis. A longitudinal incision was made along the posterior midline of the elbow joint, followed by separation of the subcutaneous fascial tissue. The dissection extended from the lateral and medial epicondyles, with careful medial dissection to expose the ulnar nerve. At the tip of the olecranon, two 2.0 mm Kirschner wires were pre-drilled parallel to the ulnar styloid. A transverse osteotomy was then performed at the mid-olecranon using a thin saw blade, with the osteotomy flipped proximally to expose the humeroradial joint. Intra-articular bruising and fracture fragments were cleared, and the fractures and ligament injuries were thoroughly assessed. The following structures were then repaired in a deep-to-superficial sequence: coronoid process, anterior joint capsule, radial head, lateral collateral ligament, and medial collateral ligament. The olecranon was repositioned and fixated using Kirschner wires and tension bands, and the ulnar nerve was returned to its anatomical location. A lateral stress test was performed intraoperatively to assess elbow joint stability. After thorough wound irrigation and ensuring optimal hemostasis, the incision was closed in layers, and a conventional drainage tube was inserted.

Treatment of important structures

Coronoid fracture

In cases with Regan-Morrey type I fractures, a hole was drilled at the proximal end of the ulna, moving posteriorly to anteriorly. No. 2 Ethibond sutures (Johnson & Johnson, New Brunswick, NJ, USA) were used to reposition the anterior joint capsule and coronoid fracture fragments, followed by lasso suturing. For Regan-Morrey type II and III fractures, where larger fracture fragments were present, repositioning was achieved, and multiple countersunk screws were employed for fixation.

Radial head fracture

All patients in this study with radial head fractures underwent open reduction and internal fixation. The radial head’s articular surface was realigned and fixated using headless cannulated screws (HCS). In cases of radial neck fractures, low-profile plates were applied for fixation, ensuring they were placed within the “safe zone” while monitoring radial head rotation around the center.

Soft tissue repair

Lateral collateral ligament (LCL) repair: For avulsions at the humeral epicondyle, anchor nails or transosseous drilling were used for fixation. When avulsions occurred along the mid-portion tear of LCL, continuous hemstitch suturing was performed using No.2 Ethibond sutures (Johnson & Johnson).

Medial collateral ligament (MCL) repair: The decision to repair the MCL complex was based on an evaluation of elbow stability following repairs of the coronoid, radial head, and LCL. Persistent intraoperative instability led to anchor repair for avulsions at the origin or insertion points, while body avulsions were addressed with continuous hemstitch sutures using No. 2 Ethibond sutures (Johnson & Johnson, New Brunswick, NJ, USA). Additionally, injuries to the flexor-pronator teres complex were repaired simultaneously when present.

Postoperative management

Postoperatively, patients were administered oral indomethacin (25 mg, three times daily) for six weeks to prevent HO. The elbow joint was immobilized at a 90° flexion angle, and after three days of neutral position plaster slab fixation, patients began passive elbow extension, flexion, and forearm rotation exercises using a hinged brace. In cases of tenuous reconstructions, forearm rotation was restricted to protect the repair. Two weeks post-surgery, patients started active flexion, extension, and forearm rotation exercises. The hinged brace was removed by the sixth postoperative week.

Follow-up and evaluation of efficacy

Patients were re-examined monthly for the first six months following surgery and every three months thereafter, with a minimum follow-up period of one year. During these follow-ups, the following data were recorded: pain levels, elbow flexion and extension range of motion, forearm rotation, elbow stability, postoperative complications, and X-ray findings. The Visual Analogue Scale (VAS) was used to assess pain both preoperatively and at the final follow-up. Additionally, the Physical Component Summary (PCS) and Mental Component Summary (MCS) scores from the SF-36 questionnaire were employed to evaluate patients’ quality of life, with calculations performed for both components. Elbow function was assessed using the Mayo Elbow Performance Score (MEPS) (Morrey & Adams, 1992), which evaluates four domains: pain (maximum score of 45), joint mobility (20), joint stability (10), and daily functions (25). MEPS scores were classified as excellent (≥90), good (75–89), acceptable (60–74), and poor (≤60).

Statistical analysis

Data analysis was conducted using SPSS 26.0 statistical software (SPSS, Inc., Chicago, IL, USA). Quantitative variables were described as means, standard deviations, and ranges, while qualitative variables were presented as frequencies and percentages. Pre- and postoperative comparisons of VAS and SF-36 scores were performed using a two-tailed paired t-test. A p-value of less than 0.05 was considered statistically significant.

Results

A total of 73 patients with TTE who underwent treatment using the olecranon osteotomy approach were identified from our database. The patients’ ages ranged from 18 to 68 years, with a mean age of 40.0 ± 15.1 years. The causes of injury were as follows: nine cases resulting from car accidents, three from falls, and six from falls from height. Radial head fractures were classified using the Mason method, with 11 cases categorized as type I, 42 as type II, and 20 as type III. Coronoid fractures were classified according to the Regan-Morrey method, with 45 cases classified as type I, 23 as type II, and five as type III. Associated injuries included five cases of distal radius fracture, three of scaphoid fracture, and three of proximal humerus fracture. The mean time between injury and surgery was 5.6 ± 1.6 days (range, 3–8 days) (Table 1).

Follow-up was conducted over a period of 15 to 60 months, with a mean duration of 37.1 ± 13.3 months. Postoperative X-rays revealed that all fractures had healed within 3 to 6 months, with an average healing time of 4.3 ± 1.1 months. By the final follow-up, all patients had regained normal elbow function. The average elbow flexion was 124.4° ± 9.2°, while the average extension was 9.6° ± 6.5°. The mean range of motion for flexion and extension was 114.8° ± 11.7°. The mean forearm pronation was 77.3° ± 4.8°, supination averaged 79.2° ± 6.5°, and the total range of forearm rotation was 156.5° ± 8.4°. The average MEPS score was 89.3 ± 6.4, with 36 patients classified as having excellent outcomes and 37 with good outcomes (Table 2).

Table 1 Patient demographics.

Patient number	73	
Age (years)	40.0  ±  15.1 (18∼68)	
Male:female	44 (60.3%):29(39.7%)	
Causes of injury		
Car accident	24	
Fall down	24	
Fall from height	25	
Left or right side		
Left	37	
Right	36	
Radial head fracture classification (Mason)		
I	11	
II	42	
III	20	
Coronoid fracture classification (Regan-Morrey)		
I	45	
II	23	
III	5	
Combined injury		
Distal radius fracture	5	
Scaphoid fracture	3	
Proximal humerus fracture	3	
Time to operation (d)	5.6  ±  1.6 (3∼8)	

Table 2 Follow-up and clinical outcomes for all patients.

Variable	Results	
Follow-up (months)	37.1  ±  13.3 (15∼60)	
Fracture healing time (months)	4.3  ±  1.1 (3∼6)	
Elbow Range of Motion (°)		
Flexion	124.4  ±  9.2 (110∼140)	
Extension	9.6  ±  6.5 (0∼20)	
ROM of flexion and extension	114.8  ±  11.7 (90∼140)	
Forearm Range of Motion (°)		
Pronation	77.3  ±  4.8 (70∼85)	
Supination	79.2  ±  6.5 (70∼90)	
ROM of rotation	156.5  ±  8.4 (140∼175)	
MEPS	89.3  ±  6.4 (80∼100)	
Excellent	36 (49.3%)	
Good	37 (50.7%)	
Acceptable	0	
Complications		
Heterotopic ossification	7 (9.6%)	
Transient ulnar nerve palsy	8 (11.0%)	

Postoperatively, seven patients developed HO, and eight experienced transient ulnar nerve palsy, which resolved within five days to three weeks following nerve nutrition therapy. No cases of persistent pain, incisional infection, elbow instability, traumatic arthritis, or delayed ulnar neuritis were observed. Table 3 outlines the results of the VAS scale and SF-36 questionnaire before surgery and at the final follow-up. The mean preoperative VAS score was 8.78 ± 1.11, which significantly improved to 0.97 ± 0.83 at the final follow-up (p = 0.000). Preoperative PCS and MCS scores from the SF-36 were 45.77 ± 3.59 and 60.67 ± 3.91, respectively, with substantial postoperative improvements to 93.85 ± 2.65 (p = 0.000) and 95.79 ± 3.11 (p = 0.000). A representative case is shown in Fig. 2.

Table 3 Comparison of preoperative VAS and SF-36 scores with their respective follow-up scores.

	Preoperative	Follow-up	p-value	
VAS	8.78  ±  1.11	0.97  ±  0.83	0.000	
SF-36				
PCS	45.77  ±  3.59	93.85  ±  2.65	0.000	
MCS	60.67  ±  3.91	95.79  ±  3.11	0.000	
Notes.

VAS Visual Analogue Scale

PCS Physical Component Summary

MCS Mental Component Summary

Figure 2 Treatment of the terrible triad of the elbow using a single olecranon osteotomy.

Typical case: a 25-year-old male patient presented with TTE resulting from a fall from a height. (A–C) Preoperative X-ray images show an elbow dislocation accompanied by coronoid and radial head fractures. (D) X-ray taken 20 months post-surgery demonstrates complete fracture healing, with normal elbow joint spacing and no complications such as joint degeneration or heterotopic ossification. (E) Clinical examination 20 months post-surgery reveals full restoration of elbow flexion, extension, and forearm rotation to normal range.

Discussion

Various surgical approaches have been documented in the literature for treating elbow fractures, including the lateral, combined mediolateral, posterior, anterior, and combined anterolateral procedures (Carroll & Morrissey, 2022; Chen & Bi, 2016; Hou et al., 2021; Meena et al., 2020; Ohl & Siboni, 2021). Among these, the lateral approach—particularly the Kocher or extensor digitorum communis (EDC) split—is frequently employed due to its ability to facilitate concurrent repair of the lateral ligamentous complex (Daniels et al., 2023). However, this approach offers limited access to the coronoid process, making it effective primarily for type I coronoid fractures, partially for type II, and for associated lateral collateral ligament injuries. For types II and III coronoid fractures, more robust fixation using screws or dedicated anatomic plates is typically required, often necessitating an additional anterior or medial approach. The medial approach (Lor, Toon & Wee, 2019), involving ulnar nerve localization and medial ligament repair along with coronoid process fixation, contrasts with the anterior approach, which is more technically demanding and allows only for coronoid fixation (Ohl & Siboni, 2021; Yang et al., 2017).

The choice of surgical approach should be tailored to the anatomical configuration of the elbow and the fracture pattern. Previous studies have advocated for the posterior approach as the primary entry for elbow joint access (Das et al., 2023). The posterior approach offers the advantage of creating a full fascial flap at the deep fascial level, minimizing injury to medial and lateral dermal nerves and the subcutaneous vascular plexus. This approach also reduces the likelihood of postoperative sensory disturbances and painful neuromas. Dowdy et al. (1995) found that the posterior median incision provides simultaneous access to both the lateral and medial structures of the elbow while reducing damage to the superficial cutaneous nerve. The olecranon osteotomy technique, historically used for intercondylar comminuted fractures of the humerus (Dumartinet-Gibaud et al., 2021), allows for clear intraoperative visualization and is relatively straightforward to perform. In our study, a posterior incision was made to expose both the medial and lateral bony and soft tissue structures of the elbow. This approach simplifies repositioning and fixation, making it an accessible method for younger, less experienced surgeons. Nevertheless, the olecranon osteotomy technique has potential drawbacks. It may increase the risk of hematoma formation in the posterior elbow joint, which could lead to a higher incidence of HO (Zhou et al., 2024). In our case series, two patients developed HO in the olecranon fossa, impairing elbow mobility and necessitating a secondary procedure for elbow release and excision of the HO. Another limitation is the risk of nonunion or malunion at the osteotomy site, which may require further surgical intervention. Although no cases of nonunion or malunion were observed at the osteotomy site in our study, this approach may not be universally applicable to all elbow fracture types.

Conservative treatment of TTE without structural reconstruction is insufficient to maintain elbow stability due to the severity of the injuries. Prolonged immobilization further risks causing joint stiffness. To address these concerns, McKee et al. (2005) developed a standard treatment protocol for TTE, which includes: (1) repairing the joint capsule or internally fixing the coronoid fracture to restore coronoid stability; (2) preserving or replacing the radial head, depending on the extent of the injury, to reestablish its stability; (3) repairing the lateral collateral ligament (LCL) and related structures; (4) repairing the medial collateral ligament (MCL) if elbow instability persists despite the previous procedures; and (5) employing a hinged external brace for supplemental fixation if conventional repair does not restore stability.

The coronoid process plays a critical role in maintaining elbow stability, bearing 52% to 65% of the ulnar coronoid stress during elbow extension. It is essential for preserving axial, posteromedial, and posterolateral rotational stability, preventing elbow dislocation (De Klerk et al., 2023). Among TTE cases, Regan-Morrey types I and II fractures are common apical fractures. Doornberg, Duijn & Ring (2006) showed that in patients with TTE, coronoid fractures typically involve about 35% of the coronoid’s height. In this study, type I fractures were treated by suturing the anterior joint capsule using a lasso technique through a drilled ulnar hole. For type II and III fractures, which involve larger fragments, countersunk screw fixation was performed from anterior to posterior. In prior studies, coronoid fracture fixation was typically applied from the dorsal to the palmar aspect of the ulna (Lee, Lim & Kim, 2020; Zha et al., 2021). However, this method often failed due to insufficient fragment grip and difficulties in controlling screw orientation. To overcome these limitations, we employed the olecranon osteotomy approach. By lifting the olecranon, the coronoid fracture fragment became directly visible, allowing for precise repositioning and fixation of the ulna from the palmar to the dorsal side. In this study, no instances of failed internal fixation for coronoid fractures occurred using this technique.

The radial head plays a vital secondary role in elbow stability by providing crucial anterior and lateral support, protecting against valgus stress, and preventing posterior instability (Diez Sanchez, Barco & Antuna, 2023). After radial head resection, the elbow’s resistance to valgus stress is reduced by 30%, as demonstrated by Ring, Jupiter & Zilberfarb (2002), who found that all patients experienced elbow re-dislocation following radial head resection. However, satisfactory outcomes were observed in cases where the radial head was preserved. This underscores the recommendation against radial head resection for radial head fractures in the treatment of TTE. In Mason type I and II fractures without comminution, fixation can be achieved using countersunk screws after realignment. For radial neck fractures, mini-plate fixation is a viable option (Rebgetz et al., 2019; Yano et al., 2023). In cases of irreparable comminuted radial head fractures, radial head replacement should be considered (Lobo-Escolar, Abellan-Miralles & Escola-Benet, 2021). Leigh & Ball (2012) compared radial head replacement with internal fixation in patients with TTE, finding no significant difference in elbow range of motion between the two groups. However, scores on the shoulder, arm, and hand disability questionnaire were higher in the replacement group, leading to the conclusion that while radial head retention is associated with higher complication and reoperation rates, it should be prioritized in younger patients.

TTE also involves injury to surrounding ligamentous tissues and the joint capsule, making soft tissue repair essential after bone fixation. The LCL complex, located on the lateral side of the elbow, is essential for stabilizing the joint and preventing rotational displacement. The LCL includes the annular ligament, radial collateral ligament, and lateral ulnar collateral ligament (LUCL), with the LUCL being the most critical. In TTE, the LUCL is often avulsed from its origin at the humeral epicondyle, less frequently at the ligament body or termination point (Tedesco et al., 2024). When avulsion occurs at the epicondyle, repair can be achieved through anchor nail fixation or transosseous drilling. During surgery, the LCL is sutured to the rotational center of the elbow while the joint is flexed at 90°, maintaining constant tension throughout movement. For avulsions within the ligament body, non-absorbable sutures are used for repair.

The MCL, located on the medial side of the elbow joint, consists of the anterior, posterior, and transverse fasciculi. Among these, the anterior fascicle plays the most significant role in resisting valgus stress (Liu et al., 2022), while the posterior fasciculi contribute to posterolateral rotational stability. Whether the MCL should be routinely repaired remains a subject of debate. Some researchers (Zha et al., 2021) argue that MCL repair is generally unnecessary, except for athletes with specific functional demands. They suggest that adding a medial incision could increase soft tissue damage, surgical trauma, and the risk of postoperative complications. In cases of intraoperative elbow instability, they propose that a hinged external brace could offer sufficient protection for the bone and soft tissue. Conversely, other studies maintain that MCL repair is a critical component of the surgical process (Jung et al., 2021). However, more recent investigations have raised questions about whether MCL repair significantly improves patient outcomes (Corbet et al., 2023; Fahs et al., 2024). In this study, we utilized the olecranon osteotomy approach, which provided sufficient exposure for a thorough evaluation of MCL injury within the joint, avoiding the need for additional medial incisions. This minimized the complexity of the procedure and reduced the risk of soft tissue complications. Nevertheless, the long-term outcomes of this approach require further follow-up to assess its effectiveness.

Table 4 Literature review.

References	Cases, n	Approach	Radial head fracture classification (Mason I/II/III)	Coronoid fracture classification (Regan-Morrey I/II/III)	Mean follow up, mths (range)	VAS	MEPS	Mean flexion -extension (degrees)	Mean pronation -supination (degrees)	Complications, n (%)	
Chen & Bi (2016)	38	Lateral + medial/Lateral/Anterior medial	7/20/11	11/17/10	15	1.2  ±  0.9	94.9  ±  8.1	109.6	115.9	8 (21.1%)	
Zhang, Tan & Kwek (2017)	13	Lateral (Kocher)/Posterior	4/6/3	8/5/0	27.7	–	85	105	115	3 (23.1%)	
Zhou et al. (2018)	60	EDC/CML	3/11/46	5/55/0	26.1 (24–30)	4.3  ±  1.2	87.9  ±  6.1	105.6	143	6 (10%)	
Mazhar et al. (2018)	44	Lateral (Kocher)/ Lateral + medial	0/3/41	26/18/0	31.7	1.7  ±  0.9	90.7  ±  9.6	114.6	118	5 (11.4%)	
Lee, Lim & Kim (2020)	24	All-arthroscopic	10/12/2	7/17/0	29.8 (24–50)	–	90.3  ±  7.2	127.7	166.5	7 (29.2%)	
Hou et al. (2021)	25	Lateral	4/17/4	5/9/11	22.9 (12–36)	–	86.9  ±  12.2	112	147	2 (8%)	
Zha et al. (2021)	109	Lateral minimal (EDC)	–	–	36.1 (6–60)	0.8  ±  1.5	92.3  ±  8.8	123	151	38 (34.9%)	
Kaneshiro et al. (2022)	12	Lateral + medial	2/3/7	9/3/0	13.5 (3–43)	–	92.2  ±  6.9	122	148	1 (8.3%)	
Carroll & Morrissey (2022)	23	Posterior (Boyd)	–	–	–	–	–	–	–	0	
Corbet et al. (2023)	50	Lateral/ Lateral + medial	3/27/20	36/12/2	24.2	0.7  ±  1.3	89.1  ±  11.9	114	137	25 (50%)	
Ormiston & Hargreaves (2024)	10	Lateral (Kocher)	–	–	225.6 (192–252)	–	88  ±  14.2	120.5	–	4 (40%)	
Present study	73	Posterior (Olecranon Osteotomy)	11/42/20	45/23/5	37.1 (15–60)	0.97  ±  0.83	89.3  ±  6.4	114.8	156.5	15 (20.5%)	
Notes.

EDC Extensor digitorum communis

CML Combined lateral and medial

This study demonstrates the efficacy of the olecranon osteotomy approach in treating TTE among 73 patients. A comparison with existing literature (Table 4) reveals consistency in our findings with previous studies regarding improved postoperative elbow function and stability. The elbow motion and MEPS scores observed align with results from Chen & Bi (2016), Zhang, Tan & Kwek (2017) and Zhou et al. (2018), despite variations in surgical techniques employed for TTE. Additionally, the long-term effectiveness of surgical interventions, as shown in studies with extended follow-up periods like Ormiston & Hargreaves (2024), is corroborated by our results. Frequent complications observed include HO and nerve-related issues, which are consistent with the broader literature. HO occurred in 9.6% of cases, typically managed conservatively, though some cases required surgical intervention due to significant functional impairment. Ulnar neuropathy was noted in 11% of patients, primarily addressed with neurolysis or ulnar nerve transposition for persistent symptoms. The approach to managing these complications was individualized, aiming to minimize further surgical procedures while promoting functional recovery. This study contributes to the literature by emphasizing the olecranon osteotomy approach, which offers improved visualization and access during complex elbow reconstructions.

Several limitations should be acknowledged. The relatively small patient cohort limits the generalizability of the findings, and a larger sample size may be necessary to further explore the potential benefits. Additionally, as a retrospective study focusing on a single technique with limited follow-up time, no comparison with alternative methods was made. Future research with a larger sample size, prospective design, control group, and standardized surgical protocols is required to validate the safety and efficacy of the olecranon osteotomy approach for treating TTE.

Conclusions

In conclusion, this study highlights the effectiveness of olecranon osteotomy in managing acute elbow triad injuries, showing substantial improvements in functional recovery and patient quality of life. The marked reduction in postoperative pain and the positive Morrey elbow function scores suggest a successful therapeutic outcome. These findings provide critical insights into surgical strategies for complex elbow injuries and establish a foundation for future research focused on refining treatment protocols. Further investigation into the long-term outcomes and potential complications of this technique is essential to enhance clinical practices and improve patient care in this challenging area.

Supplemental Information

Data S1 Raw data

Supplemental Information 2 Categorical Data

Supplemental Information 3 STROBE Statement

I would like to express my heartfelt gratitude to my family and my wife, Jie Gao, for their support and strength.

Additional Information and Declarations

Competing Interests

Author Contributions

Human Ethics

Data Availability

The authors declare there are no competing interests.

Ming Zhou performed the experiments, analyzed the data, prepared figures and/or tables, and approved the final draft.

Yuan Xue performed the experiments, analyzed the data, prepared figures and/or tables, and approved the final draft.

Xueyuan Jia performed the experiments, analyzed the data, prepared figures and/or tables, and approved the final draft.

Jianbing Wang performed the experiments, authored or reviewed drafts of the article, and approved the final draft.

Yongwei Wu performed the experiments, authored or reviewed drafts of the article, and approved the final draft.

Yunhong Ma performed the experiments, authored or reviewed drafts of the article, and approved the final draft.

Zhenzhong Sun conceived and designed the experiments, performed the experiments, authored or reviewed drafts of the article, and approved the final draft.

Yongjun Rui conceived and designed the experiments, authored or reviewed drafts of the article, and approved the final draft.

The following information was supplied relating to ethical approvals (i.e., approving body and any reference numbers):

The study was approved by the Ethics Committee of Wuxi Ninth People’s Hospital (No. LW20220017). Data were analyzed anonymously; patients approved the results by consent. All clinical investigations were conducted under the guidelines of the Declaration of Helsinki.

The following information was supplied regarding data availability:

The raw data is available in the Supplementary File.

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
