# Peer review of "Treatment of the terrible triad of the elbow by olecranon osteotomy: a retrospective cohort study"

_PeerJ, doi:10.7717/peerj.18469_

## Round 0.1 · original submission · Major Revisions

1. Please provide a clearer rationale for choosing the olecranon osteotomy approach over other surgical techniques. Emphasize the potential clinical significance and societal value of the study in improving treatment strategies for the terrible triad of the elbow (TTE).
2. Clarify whether you refer to the deep surgical approach or the cutaneous ones when discussing the single posterior approach. Provide references to support your statements regarding the challenges and limitations of various surgical approaches.
3. Please specify the level of experience of the surgical team to help assess the generalizability of the study results. Discuss more specific directions for future research, such as increasing the sample size, extending the follow-up period, and conducting a multi-center randomized controlled trial.
4. Please provide more evidence and explanation regarding the treatment of radial head fractures, particularly in cases of Mason III fractures and patients with poor bone quality.
5. Explain the rationale for administering indomethacin postoperatively to prevent heterotopic ossification.
6. Discuss any restrictions on forearm rotation or varus stresses following ligament reconstruction, and clarify whether hinged braces were recommended.
7. Please improve the figures by removing unnecessary details and focusing on the operated limb, providing enlarged images for better visualization.
8. It is better to Include a comparison of your results with other papers addressing TTE.

Reviewer 1 ·

Basic reporting

This retrospective study investigated the effectiveness of a single olecranon osteotomy in treating the terrible triad of the elbow, which involves elbow dislocation with coronoid and radial head fractures. The study included 73 patients with the terrible triad who underwent surgery between 2015 and 2022. All patients were treated with a single olecranon osteotomy. Results showed that the coronoid and radial head fractures healed in all patients with an average healing time of 4.3 months. Patients regained full elbow function and achieved good to excellent clinical outcomes based on MEPS scores. Pain significantly reduced as demonstrated by VAS scores, and patients showed improvements in their quality of life as measured by the SF-36 questionnaire. The authors concluded that a single olecranon osteotomy is a safe and effective treatment for the terrible triad of the elbow, offering promising results with a relatively easier surgical procedure.
1.The author's English needs to be polished by a professional organization.
2. The abstract highlights the olecranon osteotomy as a "relatively easier and safer surgical procedure." While it may be simpler than some other techniques, this statement oversimplifies the complexity of the procedure and ignores potential complications associated with any surgery.
3. The abstract focuses solely on the olecranon osteotomy approach without comparing it to other surgical techniques for the terrible triad of the elbow. This omission limits the reader's ability to assess the potential advantages and disadvantages of this particular method.
4. In Abstract, the conclusion claims that the terrible triad of the elbow can be "effectively treated" with a single olecranon osteotomy. This statement is overly confident, given the limitations of the study, particularly its retrospective nature and lack of a control group. The abstract should avoid making definitive claims about the effectiveness of the treatment without proper controls.
5. The introduction lacks a clear research question and fails to explicitly state the study's purpose and significance. Although it mentions limitations in existing research, it lacks a deep dive into the pros and cons of different surgical approaches and the rationale behind choosing the olecranon osteotomy approach.
6. In Introduction, the description of the research methodology is also too brief, omitting details about sample size, patient characteristics, follow-up duration, and assessment metrics. Furthermore, the introduction lacks a research hypothesis, leaving the reader unclear about the expected conclusions.
7. The study's retrospective design, small sample size, lack of a control group, and reliance on subjective assessments limit the generalizability and strength of the conclusions. Further research with a larger sample, a prospective design, a control group, and standardized surgical protocols would be necessary to confirm the effectiveness and safety of olecranon osteotomy for treating the terrible triad of the elbow.
8. While the passage mentions heterotopic ossification in seven cases, it lacks context on how this complication was managed. Were these cases treated conservatively or surgically? Knowing this information would provide valuable insight into the long-term impact of heterotopic ossification on patient outcomes.
9. The discussion focuses heavily on the olecranon osteotomy approach, highlighting its benefits without sufficiently acknowledging the potential drawbacks or comparing it to other established surgical techniques. A more balanced perspective would provide a more objective assessment of the procedure's strengths and weaknesses.
10. The discussion mentions the debate surrounding MCL repair in TTE but doesn't provide a comprehensive overview of the arguments for and against it. Presenting a more nuanced analysis of this debate would provide valuable context for the reader.
11. While the discussion suggests the need for further research with a larger sample size and longer follow-up, it doesn't provide specific recommendations for future study design or directions. Identifying specific research questions or areas of interest would enhance the discussion's impact.

Experimental design

See Basic reporting for related comments

Validity of the findings

See Basic reporting for related comments

Reviewer 2 ·

Basic reporting

1.1 The study investigates the use of a single olecranon osteotomy for treating the terrible triad of the elbow, offering a potentially less invasive alternative to traditional approaches. The paper evaluates various outcome measures, including elbow function, pain, and quality of life, providing a holistic assessment of treatment efficacy. However, the language of the manuscript can be further professionally polished to make it easier for the reader to understand.
1.2 To be sure, the references in this manuscript are very reasonable, and there are many documents in the last 3 years, which can have a better background to support the author's research.
1.3 The structure of the article is reasonable, the chart is basically clear, and the overall writing is in line with the standard.

Experimental design

2.1 The authors describe in detail the definition, clinical presentation, and challenging treatment of the elbow terror triad (TTE) in the introductory section with sufficient background information. However, the potential clinical significance and societal value of the study for improving TTE treatment strategies could be further emphasised to give the reader a clearer understanding of the importance of the study.
2.2 The study used a retrospective cohort design with clear methodological descriptions, including patient inclusion and exclusion criteria, surgical procedures, and postoperative management, to ensure the scientific validity and reproducibility of the study. It is recommended that the level of experience of the surgical team be clarified in order to assess the extrapolation of the study results.
2.3 The authors have identified limitations of the study, such as small sample size and relatively short follow-up time. More specific directions for improvement, such as enlarging the sample size, lengthening the follow-up time, and conducting a multi-centre randomised controlled trial, are suggested to enhance the scientific validity and clinical application value of the study.

Validity of the findings

3.1This study included 73 patients, a moderate sample size, but it is recommended that the authors further discuss the basis for determining the sample size and the generalisability of the findings. Also, consider whether similar data from other centres or regions are available for comparison to enhance the persuasiveness of the findings.
3.2 Regarding " All patients with fractures of the radial head in this study underwent an open reduction and internal fixation" Please provide more evidence and explanation
3.3 Regarding "The patients were orally administered indomethacin (25 mg, 3 times/d) for six weeks to avoid heterotopic ossification, postoperatively"
3.4 Detailed Discussion of Complications: Provide a more detailed analysis and discussion of the complications encountered in the study, including their incidence, severity, and management.

Reviewer 3 ·

Basic reporting

- English language review is encouraged
- Introduction should better highlight the rationale for the choice of the olecranon osteotomy;
- Structure ok
- Raw data ok
- Figures could be improved by removing unnecessary details and providing a focus on the operated limb (enlarged)

Experimental design

It should be improved the rationale
The research question is not clear: what do you want to demonstrate?

Validity of the findings

Unapplicable.

Additional comments

Lines 58-61: you should specify whether you refer to the deep surgical approach or to the cutaneous ones . It is possible by a single posterior cutaneous approach (global approach), to develop both medial and lateral deep approaches as well as Taylor and Sham or flexor carpi ulnaris split approach and Hotchkiss approach by rising cutaneous flaps.

Lines 61-62: Actually, it is possible to decide how to approach the TTI on the basis of the radial head fracture. If a non fixable RH fracture is present, the lateral approach (mainly Kocher) combined with the lesion of the LCLU allows you enough workplace to address the coronoid (anterolateral coronoid fracture that is typical of TTI); otherwise, whether a reconstructable RH fracture is present, you can address the coronoid by an Hotchkiss approach. It should be mentioned that the tip of the coronoid when the lateral column (RH) is restored, is not vital to elbow stability and it could be ignored.

Lines 65-67: please, add reference for this statement

Lines 70-71: please, better clarify your hypothesis.

Lines 107-108: a transverse olecranon osteotomy is prone to higher instability than an “en chevron” one: why did you choice it?

Lines 123-125: it seems that you used olecranon osteotomy just to improve the surgical view but it didn’t change the standard fixation method that is with screws from behind. I would be worried about introducing another fracture (iatrogenous) when I could easily see the fracture and reduce it by a medial approach.
Besides, the Morrey classification is outdated since it only allows a sagittal view classification. The O’Driscoll classification allows for a more comprehensive classification and is prognostic for the type of treatment to perform.

Lines 127-131: Were you able to fix all of the Mason III fractures? Even in poor bone quality? I wander you didn’t use a RH prosthetis. Could you confirm? You state you have patients until 68 ys of age: quite brittle bone, doesn’t it?
What do you mean by microplates? Low profile plates?

Lines 134-135: what do you mean by “humeral body”? You could classify the ligamentous lesions by the PURCCSS classification by Giannicola et al 2013

Lines 137: do you always repaired the MCL complex? This is a discussion point, since many Authors state when the elbow is stable after the coronoid, RH and LCL repair, the MCL should be addressed only if a residual instability remains. Please, comment on this choice

Lines 142-148: do you never restrict forearm rotation in case of tenuous reconstruction? What about the avoiding of varus stresses upon reconstructed ligaments? Do you recommend hinged braces ?

Lines 203: Actually, it is not true. Since the rotatory mechanism you reported in introduction, even a standing height fall can cause such injuries. High energy traumas cause a much more complex fracture pattern

Lines 203-218: this part of the discussion is unnecessary since the traumatic mechanism is not the topic of this paper

Lines 221: when you state lateral approach you should specify which interval you mean: Kocher or Kaplan? Kocher approach allows a direct repair of the ligament; such statement can’t be valid for the Kaplan approach

Lines 227-229: if you have a type III fracture (basal) you can use the flexor carpi ulnari split (FUC split) and you can easily address the fracture; it is worth of mention that in such fracture-dislocation, the anterior capsule is torn

Lines 230-231: this statement by O’Driscoll refers to the global approach that is only a posterior skin incision not a posterior deep interval (olecranon osteotomy)

Lines 255-261: You can read the work by Watts et al on the three column concept for elbow instability. You can see that the anterolateral coronoid can be ignored if enough stability is obtained

I’m surprised no complications occurred.

No comparisons between this work’s results and other papers addressing TTI is reported neither a control group with standard procedure is present, so no statement on superiority can be done or other similar conclusions.


Tables : ok

---

## Round 0.2 · accepted · Accept

Since authors have made revisions according to all comments, I think this paper can be accepted for publication.

Reviewer 1 ·

Basic reporting

I am glad that the author has made a more detailed revision, which has solved my doubts. I agree that the revised manuscript be accepted.

Experimental design

no comment

Validity of the findings

no comment

Reviewer 2 ·

Basic reporting

I have carefully reviewed the author's revision and I think it is valid. There is no further suggestion.

Experimental design

I have carefully reviewed the author's revision and I think it is valid. There is no further suggestion.

Validity of the findings

I have carefully reviewed the author's revision and I think it is valid. There is no further suggestion.